# Correlation between Serum 25-Hydroxyvitamin D Level and Peripheral Arterial Stiffness in Chronic Kidney Disease Stage 3–5 Patients

**DOI:** 10.3390/nu14122429

**Published:** 2022-06-11

**Authors:** Chung-Jen Lee, Yi-Jen Hsieh, Yu-Li Lin, Chih-Hsien Wang, Bang-Gee Hsu, Jen-Pi Tsai

**Affiliations:** 1Department of Nursing, Tzu Chi University of Science and Technology, Hualien 97010, Taiwan; gugilee@msn.com; 2Division of Nephrology, Hualien Tzu Chi Hospital, Buddhist Tzu Chi Medical Foundation, Hualien 97010, Taiwan; hij@mail.tcu.edu.tw (Y.-J.H.); nomo8931126@gmail.com (Y.-L.L.); wangch33@gmail.com (C.-H.W.); 3School of Medicine, Tzu Chi University, Hualien 97004, Taiwan; 4Division of Nephrology, Department of Internal Medicine, Dalin Tzu Chi Hospital, Buddhist Tzu Chi Medical Foundation, Chiayi 62247, Taiwan

**Keywords:** brachial-ankle pulse wave velocity, 25-hydroxyvitamin D, chronic kidney disease, peripheral artery stiffness

## Abstract

Vitamin D deficiency and high brachial-ankle pulse wave velocity (baPWV) are each independently associated with higher incidence of mortality and cardiovascular (CV) disease or CV events, respectively. This study aimed to evaluate the relationship between serum 25-hydroxyvitamin D levels and baPWV in non-dialysis patients with stage 3–5 chronic kidney disease (CKD). We enrolled 180 CKD patients. A commercial enzyme-linked immunosorbent assay was used to measure 25-hydroxyvitamin D levels. BaPWV values were measured using an automatic pulse wave analyzer. Either left or right baPWV > 18.0 m/s was considered indicative of peripheral arterial stiffness (PAS). In this study, 73 (40.6%) patients were found to have PAS. Compared to those without PAS (control group), patients with PAS were older and had higher incidence of diabetes mellitus, higher systolic and diastolic blood pressure, higher levels of intact parathyroid hormone, and C-reactive protein, and lower levels of 25-hydroxyvitamin D. Multivariate logistic regression analysis found 25-hydroxyvitamin D levels (odds ratio [OR]: 0.895, 95% confidence interval [CI] 0.828–0.968, *p* = 0.005) and old age (OR: 1.140, 95% CI 1.088–1.194, *p* < 0.001) to be independently associated with PAS in patients with stage 3–5 CKD. Lower serum 25-hydroxyvitamin D levels and older age were associated with PAS in these patients.

## 1. Introduction

Cardiovascular disease (CVD) can result from traditional risk factors such as diabetes mellitus (DM) and hypertension (HTN) as well as risk factors specific to chronic kidney disease (CKD). It has long been recognized as the main cause of adverse long-term outcomes in CKD patients [1]. Evidence has shown that the stiffening of vascular walls, which may result from dysregulation of elastin and collagen production, oxidative stress, disordered mineral metabolism, and low-grade inflammation, can increase the risk of myocardial strain, ischemia, and future CVD in CKD patients [2,3]. To assess arterial stiffness requires specific techniques while a few non-invasive methods have been developed. Among them, brachial-ankle pulse wave velocity (baPWV) is a simple, non-invasive means of determining the stiffness of large to medium-sized arteries and is a predictor of the risk of cardiovascular (CV) events and mortality in the general population and DM, HTN, and CKD patients independent of other CV risk factors [4,5,6,7]. In addition, baPWV had been found to be associated with decline of estimated glomerular filtration rate of CKD stage 3 to 5 patients as well as being an independent predictor for commencement of dialysis and death [8].

25-Hydroxyvitamin D is thought to be a reliable indicator of vitamin D levels. It has an established role in mineral homeostasis and musculoskeletal function and is also known to exert extra-skeletal effects, including modulation of endothelial function, immune function, inflammatory responses, and cell cycle regulation [9]. Deficiencies in 25-hydroxyvitamin D are associated with albuminuria [9,10] and impaired renal function in the general population [11] and all-cause mortality and kidney failure requiring long-term dialysis in pre-diabetic and DM patients [12,13]. Therefore, the reports of KDIGO (Kidney Disease: Improving Global Outcome) recommends regular measurement of serum levels to ensure sufficient vitamin D levels in patients with stage 3–5 CKD [14]. With regard to vascular function, vitamin D could affect the abilities of vasocilation through modulating the expression of vascular smooth muscle cells and the bioavailability of vasodilators [15,16], indicative of reduced nitric oxide production as well as dysregulated contents of elastin and collagens [17]. In addition, 25-hydroxyvitamin D levels are inversely associated with carotid-femoral pulse wave velocity (cfPWV) in the general population, the elderly, and newly diagnosed HTN patients [18,19,20]; with aortic intima-media thickness in subclinical atherosclerosis patients [21]; and with elevated augmentation indexes and increased levels of vascular calcification in CKD patients [22,23].

The ability of baPWV to identify arterial stiffness suggests its potential for clinical use in the classification of vascular abnormalities and as a simple, but useful, marker for use in CVD risk assessments. In light of both this and the role played by vitamin D in the modulation of vascular function and its association with adverse CV events, we aimed to examine the association between serum 25-hydroxyvitamin D levels and baPWV values in non-dialysis patients with stage 3–5 CKD.

## 2. Materials and Methods

### 2.1. Patients

Between January and December 2016, patients of the renal outpatient department of our medical center were enrolled in this study. The inclusion criteria were being older than 18 years and having an estimated glomerular filtration rate (eGFR) of >60 mL/min per 1.73 m^2^. This was calculated using the CKD-EPI (Chronic Kidney Disease Epidemiology Collaboration) equation. A patient was deemed to suffer from CKD when two separate measurements, taken at least 3 months apart, showed their eGFR to be <60 mL/min per 1.73 m^2^. CKD stage was then determined using the criteria of the Kidney Disease Outcomes Quality Initiative. Patients with an eGFR of 30–59 were classed as stage 3; those with an eGFR of 15–29 were stage 4; and those with an eGFR < 15 mL/min per 1.73 m^2^ were classed as having stage 5 CKD [14]. Diagnosis of DM was applied to patients with fasting plasma glucose higher than 126 mg/dL, and to those using oral hypoglycemic medications or insulin [24]. HTN was defined as having systolic blood pressure (SBP) ≥ 140 mmHg and/or diastolic blood pressure (DBP) ≥ 90 mmHg or having taken anti-hypertensive agents in the past 2 weeks. Patients with malignancies, inflammatory diseases, heart failure, limb amputations, or obstructive pulmonary disease, as well as those with arteriovenous shunts (grafts) and those undergoing renal replacement therapy at the time of blood sampling, were excluded from the study. Those not wishing to participate were also excluded. This study was approved by the Research Ethics Committee of Tzu Chi Hospital (IRB103-136-B).

### 2.2. Anthropometric and Biochemical Analysis

Patient variables were measured after overnight fasting. Body weight and height measurements were recorded to the nearest 0.5 kg and 0.5 cm, respectively. Body mass index was calculated as weight (kg) divided by height squared (m^2^). Blood samples (approximately 5 mL) were centrifuged immediately after collection at 3000× *g* for 10 min. Serum levels of blood urea nitrogen, creatinine, fasting glucose, total cholesterol, triglycerides, low-density lipoprotein cholesterol (LDL-C), total calcium, phosphorus, and C-reactive protein (CRP) were measured using an autoanalyzer (Siemens Advia 1800; Siemens Healthcare GmbH, Henkestr, Germany) [21,22]. Human intact parathyroid hormone (iPTH) levels (Abcam, Cambridge, MA, USA) and 25-hydroxyvitamin D levels (Crystal Chem USA, Elk Grove Village, IL, USA) were measured using commercially available enzyme-linked immunosorbent assays.

### 2.3. Measurement of Brachial-Ankle Pulse Wave Velocity

After blood sampling, patients rested for 10 min, and then baPWV was measured using a plethysmograph (VaSera VS-1000, Fukuda Denshi Co., Ltd., Tokyo, Japan) with pneumatic cuffs connected to plethysmographic and oscillo-metric sensors around both upper arms and both ankles [21,22]. The baPWV was calculated as the length of an arterial segment between the brachium and the ankle divided by the time interval between the wavefront of the brachial waveform and that of the ankle waveform. Based on previous studies, a baPWV of >18 m/s was taken to denote a high risk of CVD and HTN [25,26]. Patients with a left or right baPWV > 18.0 m/s were sorted into a PAS group, and the remaining patients formed the control group.

### 2.4. Statistical Analysis

Continuous variables were analyzed using a Kolmogorov-Smirnov test to determine normality and expressed as the mean ± standard deviation or the median and interquartile range (IQR) accordingly to non-normality. Variables expressed as mean ± SD were analyzed with a two-tailed Student’s independent *t*-test and those expressed as median and IQR with the Mann–Whitney *U* test. Categorical variables were expressed as numbers and percentages and analyzed by the χ^2^ test. Multivariable logistic regression analysis was used to determine the relationship between variables and PAS. A receiver operating characteristic (ROC) curve was used to determine the optimal cutoff value of serum 25-hydroxyvitamin D for the prediction of PAS in patients with stage 3–5 CKD. A correlation analysis between 25-dyhydroxyvitamin D and variables including age, BMI, eGFR, calcium, phosphorus, and iPTH, was performed using Pearson’s correlation or Spearman’s rank correlation coefficient according to the Kolmogorov-Smirnov test. All statistical analyses were performed using SPSS software for Windows version 19.0 (SPSS, Chicago, IL, USA). A *p*-value of <0.05 was considered statistically significant.

## 3. Results

The baseline biochemical and demographic characteristics of the 180 CKD patients, sorted into a PAS group (*n* = 73) and a control group (*n* = 107), are presented in Table 1. Patients in the PAS group were older and had a higher incidence of DM and higher SBP, DBP, fasting blood sugar, iPTH, and CRP levels but lower eGFR and 25-hydroxyvitamin D (15.96 ± 5.87 ng/mL vs. 19.65 ± 5.41 ng/mL, *p* < 0.001) levels. There were no statistically significant differences according to sex, medications used, the presence of comorbid HTN, or the stage of CKD between the two groups.

Multivariate logistic regression analysis adjusted for DM, age, SBP, DBP, fasting glucose, eGFR, iPTH, CRP, and 25-hydroxyvitamin D revealed age (odds ratio (OR) 1.140, 95% confidence interval [CI] 1.088–1.194, *p* < 0.001) and serum 25-hydroxyvitamin D level (OR 0.895, 95% CI 0.828–0.968, *p* = 0.005) to be independent predictors of PAS in CKD patients (Table 2).

Our ROC curve analysis found that the optimal cut-off value of serum 25-hydroxyvitamin D for the prediction of PAS is 20.08 ng/mL, with a sensitivity of 82.19% (95% CI 71.5%–90.2%), a specificity of 45.79% (95% CI 36.1%–55.7%), and an area under the curve of 0.684 (95% CI 0.611–0.751, *p* < 0.001) (Figure 1).

The outcomes of correlational analyses and linear regression of baPWV and clinical variables are shown in Appendix A. Left and right baPWV were positively correlated with age, SBP, DBP, and logarithmically transformed CRP (log-CRP) and were negatively correlated with serum 25-hydroxyvitamin D. Right baPWV was negatively correlated with eGFR. In multivariate stepwise linear regression analysis, age (*β* = 0.542, adjusted coefficient of determination (r^2^) = 0.197, *p* < 0.001), DBP (*β* = 0.370, adjusted r^2^ = 0.172, *p* < 0.001), and serum 25-hydroxyvitamin D (*β* = −0.297, adjusted r^2^ = 0.082, *p* < 0.001) were significantly correlated with left baPWV, whereas age (*β* = 0.550, adjusted r^2^ = 0.206, *p* < 0.001), DBP (*β* = 0.363, adjusted r^2^ = 0.158, *p* < 0.001), and serum 25-hydroxyvitamin D (*β* = −0.244, adjusted r^2^ = 0.055, *p* < 0.001) were significantly correlated with right baPWV. Both left and right baPWV were correlated with age, DBP and serum 25-hydroxyvitamin D level in non-dialysis patients with stage 3–5 CKD.

In this study, 97.2% of CKD patients had 25-hydroxyvitamin D levels < 30 ng/mL and 65% had levels below < 20 ng/mL. Table 3 shows the correlations between vitamin D levels and clinical variables. A significant correlation was found between 25-hydroxyvitamin D and BMI (*r* = −0.177, *p* = 0.018) and iPTH (*r* = −0.203, *p* = 0.006) but not between 25-hydroxyvitamin D and age, calcium, or phosphorus.

## 4. Discussion

This study found old age and low 25-hydroxyvitamin D levels to be independently correlated with baPWV values in non-dialysis patients with stage 3–5 CKD.

Multiple risk factors have been identified for irreversible anatomical arterial stiffening, which results in high vascular smooth muscle tone and raised blood pressure as well as irreversible changes to vascular wall structures [2,3]. These abnormalities increase pulse pressure and lower impedance circulation, exposing organs to high blood pressure and mechanical strain, with consequences such as CVD, renal dysfunction, and mortality [2,3]. Research suggests that aging might also contribute to structural and functional changes of vessels in CKD patients caused by abnormal mineral metabolism and resulting in elastin fragmentation and medial layer calcification [2,27]. In subjects without CVD [5,28], arterial stiffness, as measured by baPWV, is associated with age, impaired vascular distensibility, and average blood flow measured by magnetic resonance [5]. Moreover, in longitudinal studies, higher baPWV is an independent predictor of all-cause mortality and major CV-related events in patients with acute stroke [4], HTN [5,29], or acute coronary artery disease [6]. A meta-analysis of 14,673 Japanese participants without CVD showed that, apart from traditional risk factors, baPWV could independently predict future CVD with a 1.19-fold increase in CVD risk per standard deviation elevation [30]. Another meta-analysis of high risk CVD patients, in which more than half had pre-existing CVD or end-stage renal disease, showed that baPWV could predict 2.95, 5.36, and 2.45 relative risk of total CV events, CV mortality, and all-cause mortality, respectively [31]. Based on the preceding evidence and our findings, we propose that baPWV could be a useful indicator of arterial stiffness.

In this study, we chose to use baPWV to measure arterial stiffness, but in daily clinical practice, it can also be measured using cfPWV [32,33]. BaPWV integrates the properties of both central and peripheral arteries and is thought to be more representative of stress load on the left ventricle than is cfPWV [25,32]. Compared to cfPWV, baPWV correlated better with indices of the left ventricle and arterial stiffness, including left ventricular mass, isovolumic relaxation constant, carotid incremental modulus, effective arterial elastance, and carotid augmentation index [34]. Two cross-sectional multicenter studies with 954 and 2287 participants found similar positive correlations between both cfPWV and baPWV and age, SBP, and Framingham risk scores, carotid intima-media thickness, coronary artery disease, and stroke [32,35]. The accuracy and reproducibility of baPWV have been confirmed, and it is a much simpler measure, requiring only the wrapping of pressure cuffs around the four extremities. Therefore, we selected this modality to indicate the presence of PAS (baPWV > 18 m/s) in this study [25,26].

Blood levels of 25-hydroxyvitamin D are considered indicative of human vitamin D status and can be taken to represent bone health and as an independent predictor of several chronic diseases [9,36]. Evidence had shown that 25-hydroxyvitamin D deficiency is related to albuminuria [9,10], impaired renal function [11], the need for long-term dialysis, and all-cause mortality [13]. Regarding the modulation of vascular functions, in vitro studies have demonstrated that 1,25-dihydroxyvitamin D can affect the proliferation of vascular smooth muscle cells and impair endothelium-dependent vasodilation because of the decreased bioavailability of vasodilators [15,16]. Moreover, vitamin D receptor knockout mice were found to have reduced nitric oxide production, resulting in dysregulation of intact vascular function with increased pulse pressure, aortic augmentation pressure and index, while aortic tissue showed a marked increase in collagen contents and a decrease in elastin contents, indicative of arterial stiffening [17]. The Baltimore study of aging found an inverse association (adjusted r^2^ = 0.34, beta = −0.34, *p* = 0.04) between cfPWV and 25-hydroxyvitamin D in healthy participants, irrespective of traditional risk factors [20]. Another study similarly showed that subjects in the lowest quartile for 25-hydroxyvitamin D levels (<20 ng/mL) had the highest aortic pulse wave velocity (9.04 m/s) with 2.04-fold higher risk of having aortic stiffness [14]; HTN patients with the lowest 25-hydroxyvitamin D levels had the highest cfPWV, high-sensitivity C-reactive protein (hsCRP) levels, and left ventricular mass indexes with correlation coefficient −0.432, −0.143, and −0.235, respectively [19]. Transesophageal echocardiograms have revealed an independent inverse association between 25-hydroxyvitamin D levels and thoracic aortic intima-media thickness as well as highly sensitive CRP in subclinical atherosclerosis patients [21]. Akdam et al. found that 94.1% of patients with advanced CKD had 25-hydroxyvitamin D levels less than 30 ng/mL and there was a tendency to decrease as the CKD stage increased [22]. Moreover, there was an inverse association between 25-hydroxyvitamin D levels and augmentation indexes as well as non-significantly higher iPTH level [22]. Garcia-Canton et al. found that 18.5% of stage 4–5 CKD patients had adequate 25-hydroxyvitamin D levels (>30 ng/mL) and that lower 25-hydroxyvitamin D was associated with higher vascular calcification scores, determined from plain X-ray images [23]. Our results showed there was negative relationship between 25-dihydroxyvitamin D and iPTH and evidence had shown that down-regulation of vitamin D could up-regulate the production of iPTH, which might lead to arterial hypertension and adverse vascular remodeling and resulting in vascular wall calcification [14,22,37]. Moreover, in this study, patients in the PAS group were found to have higher iPTH than control group, and we considered that level of iPTH could play a role in vascular calcification but showed no significant association. Nevertheless, we supposed iPTH could modulate the association of PWV and 25-hydroxyvitamine D, but did not alter the results of the relationship between baPWV and 25-dydroxyvitamin D by multivariate analysis. Furthermore, evidence has shown chronic inflammation as a marker of CVD, and vitamin D played a crucial role in immune modulation, which reflected chronic low-grade inflammation on vasculature [21,22]. In this study, we similarly found elevated CRP in PAS group but showed no significant association with arterial stiffness after adjustment. We suggested that this might indicate that elevated CRP or iPTH could induce vascular calcification by themselves, or only as a compensatory result of down production of 25-dydroxyvitamin D, which needs more investigation. After all, our study showed a markedly higher incidence of vitamin D deficiency in stage 3–5 CKD patients, and those patients with lower levels of 25-hydroxyvitamin D had higher baPWV, which is indicative of arterial stiffness.

This study had some limitations. First, it had a cross-sectional design and a limited number of CKD patients, with 56.1% participants older than 65-year-old, and was conducted at a single center. Second, we did not measure both baPWV and cfPWV for comparison. Third, different diets and the use of dietary supplements affect levels of 25-dihydroxyvitamin D, and this was not controlled for. Thus, the current study could not extrapolate to a younger CKD population and the causal relationship between serum 25-dihydroxyvitamin D levels and PAS should be investigated more fully with more patients in a longitudinal study with controlled diet and nutritional supplement intake to clarify the effects of 25-dihydroxyvitamin D on vascular stiffness, which should be measured using both cfPWV and baPWV.

## 5. Conclusions

This study has demonstrated that, along with older age, serum 25-dihydroxyvitamin D levels are a biomarker for PAS in patients with stage 3–5 CKD. This indicates that 25-dihydroxyvitamin D could play a role in the progress of PAS and further study is needed to elucidate the mechanism involved.

## Figures and Tables

**Figure 1 nutrients-14-02429-f001:**
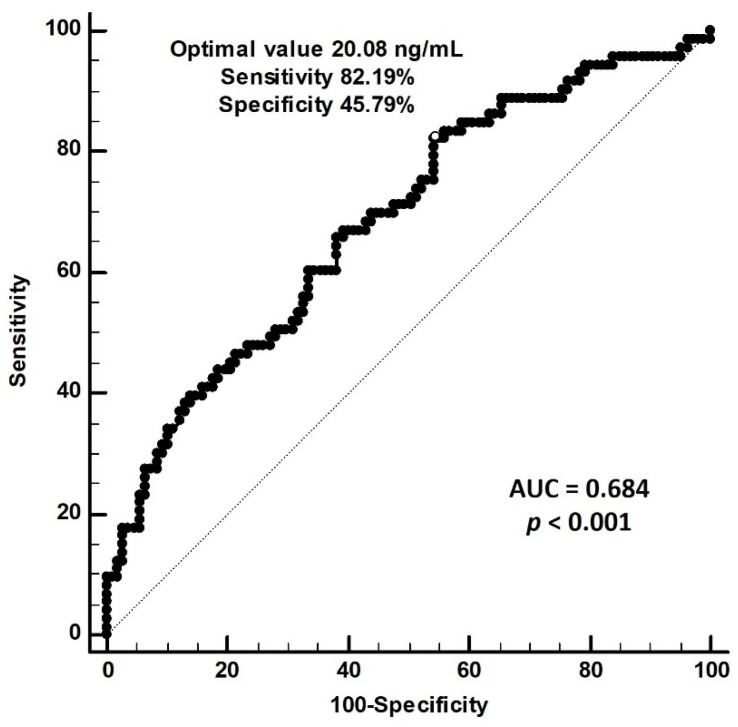
**Receiver operating characteristic (ROC) curve analysis to predict peripheral arterial stiffness in 180 chronic kidney disease patients**. The area under the ROC curve indicates the diagnostic power of serum 25-hydroxyvitamin D levels for predicting peripheral arterial stiffness among 180 chronic kidney disease patients. The AUC for 25-hydroxyvitamin D was 0.684 (95% confidence interval: 0.611–0.751, *p* < 0.001).

**Table 1 nutrients-14-02429-t001:** Clinical variables of the 180 chronic kidney disease patients with or without peripheral arterial stiffness.

Characteristics	All Patients (*n* = 180)	Control Group (*n* = 107)	PAS Group (*n* = 73)	*p* Value
Age (years)	68.39 ± 14.30	63.11 ± 13.89	76.12 ± 11.05	<0.001 *
Height (cm)	159.12 ± 8.82	159.99 ± 7.96	157.84 ± 9.86	0.110
Female, *n* (%)	81 (45.0)	43 (40.2)	38 (52.1)	0.116
Blood urea nitrogen (mg/dL)	40.44 (25.25–48.75)	40.96 (25.00–49.00)	39.68 (26.00–48.00)	0.742
Creatinine (mg/dL)	2.56 (1.50–2.80)	2.47 (1.40–2.70)	2.70 (1.60–3.25)	0.409
eGFR (mL/min)	30.45 ± 15.25	32.83 ± 15.37	26.96 ± 14.47	0.011 *
CKD stage 3, *n* (%)	82 (45.6)	55 (51.4)	27 (37.0)	0.129
CKD stage 4, *n* (%)	63 (35.0)	35 (32.7)	28 (38.4)	
CKD stage 5, *n* (%)	35 (19.4)	17 (15.9)	18 (24.6)	
Diabetes mellitus, *n* (%)	65 (36.1)	32 (29.9)	33 (45.2)	0.036 *
Hypertension, *n* (%)	148 (82.2)	87 (81.3)	61 (83.6)	0.698
Body weight (kg)	66.01 ± 14.41	67.23 ± 14.26	64.22 ± 14.53	0.169
Body mass index (kg/m^2^)	25.94 ± 4.52	26.16 ± 4.68	25.62 ± 4.29	0.435
Left baPWV (m/s)	17.39 ± 3.83	14.89 ± 1.86	21.06 ± 2.91	<0.001 *
Right baPWV (m/s)	17.29 ± 3.77	14.89 ± 1.91	20.79 ± 3.01	<0.001 *
SBP (mmHg)	149.91 ± 25.52	141.55 ± 23.57	160.68 ± 24.55	<0.001 *
DBP (mmHg)	84.49 ± 14.47	82.34 ± 13.49	87.66 ± 15.34	0.015 *
Total cholesterol (mg/dL)	159.87 ± 43.59	158.83 ± 45.76	161.38 ± 40.46	0.701
Triglyceride (mg/dL)	137.47 (83.25–166.25)	138.35 (82.00–172.00)	136.19 (83.50–157.50)	0.973
LDL-C (mg/dL)	90.24 ± 36.87	90.71 ± 38.47	89.55 ± 34.64	0.836
Fasting glucose (mg/dL)	117.40 (93.00–127.75)	113.03 (92.00–120.00)	123.81 (94.50–139.00)	0.015 *
Total calcium (mg/dL)	8.97 ± 0.57	8.95 ± 0.54	9.01 ± 0.62	0.495
Phosphorus (mg/dL)	3.68 ± 0.72	3.63 ± 0.62	3.76 ± 0.85	0.216
25-hydroxyvitamin D (ng/mL)	18.15 ± 5.87	19.65 ± 5.41	15.96 ± 5.87	<0.001 *
iPTH (pg/mL)	47.14 (30.58–54.80)	43.02 (27.00–53.30)	53.18 (36.65–60.00)	0.009 *
CRP (mg/dL)	0.81 (0.08–0.98)	0.51 (0.06–0.79)	1.25 (0.17–1.25)	<0.001 *
Current smoking, *n* (%)	20 (11.1)	12 (11.2)	8 (11.0)	0.957
ARB use, *n* (%)	112 (62.2)	68 (63.6)	44 (60.3)	0.656
β-blocker use, *n* (%)	54 (30.0)	29 (27.1)	25 (34.2)	0.304
CCB use, *n* (%)	80 (44.4)	46 (43.0)	34 (46.6)	0.635
α-adrenergic blocker use, *n* (%)	26 (14.4)	15 (14.0)	11 (15.1)	0.844
Statin use, *n* (%)	80 (44.4)	45 (42.1)	35 (47.9)	0.435
Fibrate use, *n* (%)	31 (17.2)	21 (19.6)	10 (13.7)	0.301

Values for continuous variables are given as mean ± standard deviation and tested by Student’s *t*-test; variables not normally distributed are given as median and interquartile range and tested by Mann–Whitney U test; values are presented as number (%) and analysis was done using the chi-square test. PAS, peripheral arterial stiffness; baPWV, brachial-ankle pulse wave velocity; SBP, systolic blood pressure; DBP, diastolic blood pressure; LDL-C, low-density lipoprotein cholesterol; eGFR, estimated glomerular filtration rate; iPTH, intact parathyroid hormone; CRP, C-reactive protein; ARB, angiotensin-receptor blocker; CCB, calcium-channel blocker; CKD, chronic kidney disease. * *p* < 0.05 was considered statistically significant.

**Table 2 nutrients-14-02429-t002:** Multivariable logistic regression analysis of the factors correlated to peripheral arterial stiffness among 180 chronic kidney disease patients.

Variables	Odds Ratio	95% Confidence Interval	*p* Value
25-hydroxyvitamin D, 1 ng/mL	0.895	0.828–0.968	0.005 *
Age, 1 year	1.140	1.088–1.194	<0.001 *
Diabetes mellitus, present	2.445	0.886–6.752	0.084
Systolic blood pressure, 1 mmHg	1.017	0.989–1.046	0.231
Diastolic blood pressure, 1 mmHg	1.048	0.994–1.104	0.080
Fasting glucose, 1 mg/dL	1.001	0.991–1.011	0.832
Estimated glomerular filtration rate, 1 mL/min	0.987	0.954–1.020	0.434
Intact parathyroid hormone, 1 pg/mL	1.009	0.984–1.034	0.479
C-reactive protein, 1 mg/dL	1.444	0.885–2.356	0.142

Analysis of data was carried out2 using the multivariate logistic regression analysis (adapted factors: diabetes mellitus, age, systolic blood pressure, diastolic blood pressure, fasting glucose, estimated glomerular filtration rate, intact parathyroid hormone, C-reactive protein and 25-hydroxyvitamin D). * *p* < 0.05 was considered statistically significant.

**Table 3 nutrients-14-02429-t003:** Correlation between 25-hydroxyvitamin D and clinical variables.

Correlation *p* Value	25-Hydroxyvitamin D	Age	BMI	eGFR	Calcium	Phosphorus	iPTH ^#^
**25-hydroxyvitamin D**		−0.011	−0.177	0.138	0.052	0.020	−0.203
		0.880	0.018 *	0.065	0.485	0.786	0.006 *
**Age**	−0.011		−0.035	−0.040	−0.014	−0.022	0.044
	0.880		0.644	0.592	0.850	0.774	0.554
**BMI**	−0.177	−0.035		0.042	0.017	−0.089	−0.029
	0.018 *	0.644		0.580	0.817	0.237	0.701
**eGFR**	0.138	−0.040	0.042		0.142	−0.490	−0.593
	0.065	0.592	0.580		0.058	<0.001 *	<0.001 *
**Calcium**	0.052	−0.014	0.017	0.142		0.008	−0.115
	0.485	0.850	0.817	0.058		0.916	0.124
**Phosphorus**	0.020	−0.022	−0.089	−0.490	0.008		0.318
	0.786	0.774	0.237	<0.001 *	0.916		<0.001 *
**iPTH ^#^**	−0.203	0.044	−0.029	−0.593	−0.115	0.318	
	0.006 *	0.554	0.701	<0.001 *	0.124	<0.001 *	

Analysis of data was performed using the Pearson correlation or Spearman rank correlation analysis (marked as ^#^). BMI, body mass index; eGFR, estimated glomerular filtration rate; iPTH, intact parathyroid hormone. * *p* < 0.05 was considered statistically significant.

## Data Availability

The data presented in this study are available on request from the corresponding author.

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
