# Peer review of "Correlation between Serum 25-Hydroxyvitamin D Level and Peripheral Arterial Stiffness in Chronic Kidney Disease Stage 3–5 Patients"

_nutrients, 2022, doi:10.3390/nu14122429_

Round 1

Reviewer 1 Report

The manuscript investigated the association between serum 25-hydroxyvitamin D levels and baPWV values in non-dialysis patients with stage 3–5 CKD. The manuscript is suitable for publication after minor revision. Why the authors focused on the old age and low 25-hydroxyvitamin D levels, give a reasonable reply. The introduction part is very short and it should be contained more facts and a literature review. The data obtained in Figure 1 should be discussed in depth. The authors should rewrite the conclusion part to have more quantitative data.

Author Response

The manuscript investigated the association between serum 25-hydroxyvitamin D levels and baPWV values in non-dialysis patients with stage 3–5 CKD. The manuscript is suitable for publication after minor revision.

Why the authors focused on the old age and low 25-hydroxyvitamin D levels, give a reasonable reply.

Ans: Thanks for your comments. Between January and December 2016, we enrolled CKD patients without exclusion criteria. Patients found to be old age and have low 25-hydroxyvitamin D were considered to be their baseline characteristics. However, we would mention this point in the limitation paragraph in the Discussion section as that the results could not extrapolate to younger population.

The introduction part is very short and it should be contained more facts and a literature review. The data obtained in Figure 1 should be discussed in depth. The authors should rewrite the conclusion part to have more quantitative data.

Ans: Thanks for your comments. We revised the contents in Introduction and Discussion sections.

Reviewer 2 Report

The authors analyzed serum 25- hydroxyvitamin D and brachial-ankle pulse wave velocity.

 The topic of this study is interesting, and the authors wrote a well-structured manuscript to show the result. The method of statistics is plausible.

 I have a few comments to improve the manuscript.

They used Pearson’s correlation or Spearman’s rank correlation coefficient. However, they transformed non-normally distributed continuous variables logarithmically. I wonder if they used Spearman's rank correlation coefficient or logarithm transformation in non-normally distributed continuous variables. It seems that they only used logarithm transformation.

In addition, in Table 3 did the author use log-iPTH?

The manuscript is well-written, however, I am afraid that the topic of this study is a little bit out of this journal’s scope. Although vitamin D is important as one nutrient, this manuscript is suitable for journals for general topics or focusing on the kidney field.

Therefore, I would like to ask the authors to revise the manuscript to emphasize vitamin D or nutritious aspects.

Author Response

They used Pearson’s correlation or Spearman’s rank correlation coefficient. However, they transformed non-normally distributed continuous variables logarithmically. I wonder if they used Spearman's rank correlation coefficient or logarithm transformation in non-normally distributed continuous variables. It seems that they only used logarithm transformation.

In addition, in Table 3 did the author use log-iPTH?

 Ans: Thanks for your comments. A correlation analysis between 25-dyhydroxyvitamin D and variables including age, BMI, eGFR, calcium, phosphorus, and iPTH, was performed using Pearson’s correlation or Spearman’s rank correlation coefficient according to the Kolmogorov-Smirnov test (Table 3). The description of correlation analysis in the Statistic analysis Section was revised.

Therefore, we use Spearman rank correlation analysis instead of using logarithmically transformed variables (iPTH) for analysis in Table 3.

The manuscript is well-written, however, I am afraid that the topic of this study is a little bit out of this journal’s scope. Although vitamin D is important as one nutrient, this manuscript is suitable for journals for general topics or focusing on the kidney field.

Therefore, I would like to ask the authors to revise the manuscript to emphasize vitamin D or nutritious aspects.

Ans: Thanks for your comments. We revised the title as “Correlation between serum 25-hydroxyvitamin D level and peripheral arterial stiffness in chronic kidney disease stage 3-5 patients”.
